# Ethics of Bariatric Surgery in Adolescence and Its Implications for Clinical Practice

**DOI:** 10.3390/ijerph20021232

**Published:** 2023-01-10

**Authors:** Valentina Martinelli, Simran Singh, Pierluigi Politi, Riccardo Caccialanza, Andrea Peri, Andrea Pietrabissa, Matteo Chiappedi

**Affiliations:** 1Department of Brain and Behavioral Sciences, University of Pavia, 27100 Pavia, Italy; 2Medway Hospital, Windmill Rd, Gillingham ME7 5NY, UK; 3Clinical Nutrition and Dietetics Unit, IRCCS Policlinico San Matteo Foundation, 27100 Pavia, Italy; 4Department of Surgery, IRCCS Policlinico San Matteo Foundation, 27100 Pavia, Italy; 5Department of Surgery, University of Pavia, 27100 Pavia, Italy; 6Vigevano Child Neurology and Psychiatry Unit, ASST Pavia, 27100 Pavia, Italy

**Keywords:** adolescent obesity, pediatric bariatric surgery, sleeve gastrectomy, ethics, psychosocial outcomes

## Abstract

Obesity is increasingly prevalent among adolescents. Clinical and research data support the use of bariatric surgery (BS) as a treatment option for severely obese adolescents, with good results in terms of weight loss, improvement or resolution of comorbidities, and compliance to follow up. Nevertheless, concerns still remain, with significant disparities among countries and ethical concerns mainly raised by performing an irreversible and invasive procedure in adolescence, with potential life-long alterations. In this context, the purpose of this narrative review was to discuss the main current ethical challenges in performing BS in adolescence and to inform appropriate clinical management in the field. The core ethical principles of autonomy, beneficence, nonmaleficence, and justice were revised in terms of patient-centered healthcare through the lens of psychosocial implications. The review concludes with a discussion regarding the potential directives for future research for effective, patient-centered, and ethical management of obesity in the adolescent population.

## 1. Introduction

Obesity is a major worldwide health problem. Epidemiological data report increasing rates of overweight and obesity among children and adolescents aged 5–19, from 4% in 1975 to over 18% in 2016 [1]. The dramatic lifestyle changes imposed by the SARS-CoV-2 pandemic led to a significant increase in obesity and eating disorders in this age group due to higher consumption of hypercaloric junk foods and a reduction in physical activity, often associated with an altered sleep–wake cycle [2]. Pediatric obesity is a multifaceted disease that can impact physical and mental health, a complex condition that interweaves biological, developmental, environmental, behavioral, and genetic factors. Excess adipose tissue can trigger many metabolic and immunological pathways, leading to serious co-morbidities, such as impaired glucose tolerance and type 2 diabetes, dyslipidemia, arterial hypertension, hyperuricemia, obstructive sleep apnea, polycystic ovary syndrome or non-alcoholic fatty liver disease [3]. Moreover, adolescents with obesity are at increased risk of psychological disturbances, including mood, emotional and neurocognitive alterations [4]. 

The treatment of obesity and metabolic syndrome in children and adolescents is focused on dietary counseling, physical activity, and behavioral changes. However, evidence showed low to moderate effects of these approaches [5]. Moreover, approved pharmacological treatments for obesity in pediatric age are limited [3,5].

In this context, BS gained growing attention as a treatment option for pediatric obesity and is currently performed at an increasing rate in severely obese adolescents who do not respond to conservative treatment [3,5,6]. Vertical sleeve gastrectomy (VSG) has become the most commonly used and recommended surgical procedure in this age group [7,8]. 

According to the American Society of Metabolic and Bariatric Surgery (ASMBS) guidelines, BS is proposed in adolescents with BMI ≥ 35 kg/m^2^ (moderate obesity) with major comorbidities or with a BMI ≥ 40 kg/m^2^ (severe obesity) with minor comorbidities [5,6,9]. International guidelines clearly highlight the importance of a multidisciplinary approach for preoperative evaluation and postoperative support [10,11] The ASMBS guidelines also emphasize the importance of understanding the social environment of the eligible patients and taking the opportunity to educate the guardians or parents about what postoperative life will entail [5]. Main contraindications include obesity that can be treated with medical therapy, substance abuse within the past year, current or planned pregnancy within 12–18 months of the surgical procedure, concomitant eating disorders, and medical, psychiatric, and psychosocial issues interfering with postsurgical recommendations and required lifestyle modification [5]. 

BS is associated with effective and sustained weight loss, comorbidities resolution, and improved quality of life [6,12]. However, the use of bariatric surgery in adolescent patients is still limited, with significant disparities among countries. Reasons include ethical concerns raised by performing an irreversible and invasive procedure in adolescence, with potential life-long alterations [13]. Within this framework, it is important to stress that adolescents may represent a vulnerable population with regard to autonomy in decision-making. Moreover, adolescents with obesity often suffer from body dissatisfaction, low self-esteem, and mood disturbances in combination with a history of eating disorder symptoms. All these have a potential impact on the decision to undergo BS and on the outcome [14]. In this regard, data about the long-term psychosocial fallout of BS in adolescents are still limited and warrant specific attention and competencies [15]. Of interest, although there is a general agreement on clinical guidelines for eligibility to BS, no standardized protocol was developed for screening evaluations, despite the obvious importance of a psychosocial evaluation before the intervention and the ethical issues posed by this kind of procedure. The primary aim of the present narrative review was to discuss the main current ethical challenges in performing BS in adolescence. The secondary purpose was to inform clinical practice in the field.

## 2. Materials and Methods

The present paper is a narrative review aimed at discussing the main current ethical issues in performing BS in adolescence. Studies were retrieved starting from a search in Medline and Scopus databases with the following keywords: adolescent obesity, adolescent bariatric surgery, pediatric metabolic surgery, sleeve gastrectomy, ethics, obesity, psychosocial outcomes, using MeSH headings. The search was limited to papers published from December 2002 to December 2021 and written in English, Italian, and French. Relevant papers cited in a retrieved paper were collected and assessed for their possible utility for this review. Two authors (V.M. and S.S.) independently screened articles by title and abstract to identify the most relevant published studies (original papers and reviews) addressing this topic. Selected papers were critically revised, and any disagreement was solved through discussion with a third author (M.C.) to reach a consensus.

All the co-authors reviewed and discussed the resulting draft according to their specific field of expertise to provide a theoretical point of view with respect to the ethical principles of autonomy, beneficence, nonmaleficence, and justice. The final version of the manuscript was then recirculated and approved by all the co-authors.

## 3. Results

### 3.1. Autonomy

Autonomy refers to the individual’s right to self-determination and to his/her ability to make decisions about his/her medical treatment based on his/her own beliefs and values. It represents the foundation for informed consent. Autonomy implies three main conditions: intentionality (i.e., the willingness to undergo that procedure), understanding (i.e., the ability to understand what will happen, in terms of the procedure and of its expected, possible, or probable consequences, both beneficial and harmful), and non-control (i.e., the absence of undue influence or coercion) [16,17]. These aspects need careful attention and assessment in adolescents, particularly regarding the decision to undergo surgical intervention for severe obesity. 

In most countries, individuals older than 12 years are considered to have medical decision-making capacity [18]. This means that they are thought to be able to understand and participate in medical decision-making, eventually with additional support from adult caregivers, unless they are proven incapable of making the best decision for themselves [18,19]. Given these premises, the psychosocial evaluation maintains a crucial role in assessing the child’s capacity to assent to a procedure that causes long-term alterations in metabolism and the exclusion of potential parental coercion or pressure [19]. Moreover, the stigma linked to obesity in Western society is another crucial factor. The body and body image development throughout childhood and adolescence represent essential aspects of one’s identity. Adolescents with obesity often suffer from body dissatisfaction, low self-esteem, teasing, and symptoms of mood deflection. Obese children are at greater risk of school dropout, bullying, and lacking social opportunities [14,20]. All these conditions may at least partially affect the decision to undergo a surgical procedure. 

Regarding understanding, the root of autonomy and consent is based on the fact that adequate and substantial information is provided to the patient about the procedure to be performed. It is important to discern if a patient and his/her caretakers truly fathom the changes that will follow post-surgery and the kind of implications the surgery could have on his/her life in the future, especially given that he/she is at a cusp in their lives where a lot of milestones await him/her, ranging from the transition to adulthood to choices regarding education, job, social and family life. Bariatric surgery has a possible impact on all of these.

A relatively recent pilot study (2019) showed the feasibility and potential usefulness of a Patient Decision Aid (PDA) instrument specifically designed for adolescents to promote shared decision-making about the treatment of severe obesity. The PDA included a brochure that described the indications, risks, and benefits of intensive lifestyle management versus bariatric surgery and lifestyle management [21]. The authors described a model of effective communication that may contribute to adequate autonomy in this age group [21]. 

One major issue in the psychosocial evaluation of adolescents seeking BS regards intellectual and developmental disabilities, which are usually found under the DSM-5-TR category named “Intellectual Developmental Disorder” (IDD), the current evolution of the previous concepts of Mental Retardation and Intellectual Disability [22]. Adolescents with IDD are nearly twice as likely to be obese than their peers. IDD, defined by significant limitations in both intellectual functioning (learning, reasoning, and problem-solving) and adaptive behavior (practical life skills) [23], may represent a contraindication to BS: it can reduce the individual’s ability to provide consent and lower adherence to pre or postoperative requirements, in turn limiting weight loss and increasing the risk of adverse events [19,24]. In this respect, data regarding the outcome of BS in patients with IDD are still limited; this is even more true when IDD is comorbid with other Neurodevelopmental Disorders, such as Autism Spectrum Disorder, a clinical condition that could further limit the individual’s ability to understand treatment goals and to cooperate effectively [25]. Despite these concerns, among adolescents with mixed etiologies of IDD who underwent sleeve gastrectomy, BS resulted in optimal weight loss at early follow-up [26], and IDD did not impact weight loss or adverse events up to two years postoperatively [27], but further studies are needed to confirm sustained benefits in the long term [25]. 

A careful, dedicated psychosocial assessment is essential to explore all these issues. Of interest, more recently, Moore et al. proposed a framework for how to conduct a robust, ethically grounded evaluation of pediatric patients presenting for BS to assess the eligibility of adolescent candidates in order to alleviate the burden of stigma and biases that may subconsciously permeate the decision-making process [19].

### 3.2. Beneficence

Beneficence is the ethical principle stating that physicians must act for the benefit of the patient, seeking ways to restore health and promote well-being [28]. A growing body of evidence proved Bariatric Surgery to be safe in children and adolescents and to result in durable weight loss and improvement of comorbid conditions [6]. Of interest, it was reported that weight loss and comorbidity resolution in adolescents tend to be better than in adults [29,30].

Laparoscopic sleeve gastrectomy (LSG), Roux-en-Y gastric bypass (RYGB), and adjustable gastric banding (AGB) are the main surgical options utilized in adolescent bariatric surgery [5]. Vertical sleeve gastrectomy (VSG) currently represents the most commonly performed procedure in this age group, given its safety, efficacy, and tolerability profile. Good results were reported regarding sustained weight loss, improvement or resolution of comorbidities, and compliance at follow-up [3,5,7,8]. It also has significant advantages in terms of metabolic effects compared to the other techniques [31]. Several longitudinal studies demonstrated 38–83% excess weight loss following VSG compared to non-surgical interventions alone [3,29,32].

Data on long-term outcomes in adolescents are still limited, and most of the studies focus on RYGB. Of interest, a recent retrospective analysis of prospectively collected data from 31 patients (mean age 16.19 ± 1.07 years) who underwent VSG between January 2008 and July 2014 investigated the surgery outcomes more than 10 years after the intervention. The authors reported a reduction of 33.2% in BMI ten years after surgery, with 21 patients (67.7%) achieving a BMI < 30 kg/m^2^, therefore resolving obesity [33]. Despite these promising results, limitations of this study included the small sample size and the rate of weight loss in follow-up patients (32%), which is common in this context but significantly reduces the generalizability of the results and their contribution to clinical management [33].

A limited number of studies investigated the psychosocial effects of BS in adolescents, with small sample sizes and short follow-ups. One of the earliest systematic reviews focused on psychosocial status and mental health among adolescents before and after BS was performed in 2014 [34]. Findings documented depressive symptoms in 15% to almost 70% of adolescents seeking BS, while anxiety and eating disorder symptoms were reported in 15–33% and 48–70% of BS candidates, respectively. With regard to post-operative psychosocial status and mental health, depressive symptoms either remitted or improved in the first 12 months after bariatric surgery. However, the review highlighted the need for larger sample sizes to define the prevalence of psychopathology more accurately. Moreover, the authors stressed the need to obtain more detailed data on the sociodemographic and ethnic characteristics as well as the patients’ family backgrounds, given the potential role of these factors in determining psychopathological symptoms and coping abilities [34].

More recently, a systematic review and meta-analysis on psychosocial outcomes following adolescent Bariatric Surgery [35] found a significant improvement in quality of life (Qol) within a follow-up range of 9–94 months across surgical procedures. Trends in Qol improvement showed the greatest increase at 12 months, with significant improvement sustained at the longest follow-up (60+ months) [35].

Long-term data for psychosocial outcomes reflecting more detailed mental health evaluations are still limited. Jarvholm et al. investigated the psychological outcomes in a cohort of Swedish adolescents aged 13–18 and found a significant decline in anxiety, depression, anger, and disruptive behavior at two years. Additionally, self-esteem and self-concept improved significantly at one-year post surgery, with stabilization seen at the two-year follow-up [36]. Hunsaker et al., instead, found a less dramatic improvement at two-year follow-up, with most subjects maintaining their symptomatic psychological status; however, remission or improvement was more common than the occurrence of new symptomatology. Of note, the latter study had a follow-up period that lasted as long as six years, making it one of the longest-term studies currently available for analysis [37]. 

More data are needed to clearly investigate mental health changes, with long-term assessments and objective-standardized protocols, though initial results in adolescent studies suggest evidence for the benefits of bariatric surgery on psychological endpoints [38]. 

### 3.3. Nonmaleficence

The obligation not to cause harm (“primum non nocere”) implies a careful understanding and acknowledgment of the risks inherent in the surgical approach and of the potential complications, coupled with the activation of all reasonable strategies to reduce them [16,28]. This topic is particularly relevant in the context of adolescent bariatric surgery and has represented one of the major arguments against the use of BS in adolescence in the past decades [28]. 

A retrospective analysis from the Metabolic and Bariatric Surgery Accreditation and Quality Improvement Program (MBASQIP) database (USA) confirmed that both Laparoscopic Sleeve Gastrectomy (LSG) and RYGB are relatively safe, but LSG is associated with a significantly lower rate of major complications in the first month after surgery [39] and with shorter operative times [40].

The safety of LSG, together with high survival rates, have been widely demonstrated, and LSG has become the surgery procedure of first choice in patients with severe obesity worldwide, both in adulthood and pediatric age. In the adolescent population, LSG appears to have a better safety profile than other bariatric operations [41].

Data on short- and middle-term outcomes in pediatrics show no major complications and a low rate of minor complications (4.3% according to Alqahtani et al.) [42] after LSG, with no evidence of mortality [3,43,44]. Potential surgical complications in the immediate postoperative period include nausea, vomiting, dehydration, anastomotic leak, and gastric tube twist, as well as volvulus; wound infection at the trocar site is also a recurrent complication described in patients with severe obesity (0.6%). In middle-term outcome, major complications are described in adults, but not recorded in pediatrics [3,42]. According to a recent retrospective analysis including 31 adolescent patients, followed up for more than 10 years after surgery, frequent long-term side effects were alopecia (48.39%) and gastrointestinal reflux disease (GERD; 22.58%). Symptomatic cholelithiasis necessitated cholecystectomy in 22.58% of the patients [33]. Long-term follow-up is recommended for gastroesophageal reflux disease secondary to the risk of developing an esophageal disease, such as Barrett’s esophagus [3]. 

With regard to psychological and psychiatric risks associated with bariatric surgery, epidemiological and research data suggested a higher prevalence of suicidal ideation and behaviors in adult patients undergoing bariatric surgery, both pre- and post-treatment [45,46,47]. Low self-esteem, depression, and suicidal behaviors are observed in this population [15,48]. Overall, 25% of adolescent patients seeking bariatric surgery reported having engaged in self-harm [49]. 

Bariatric surgery is not, fundamentally, a procedure to address chronic mental health conditions [35]. In a two-year follow-up of psychopathology prevalence and correlates among adolescent bariatric patients and nonsurgical controls, Hunsaker and colleagues reported that half of the patients with baseline disease saw symptoms improve while the other half exhibited persistent symptomatology [37]. In this study, approximately one in three adolescents had significant psychopathological symptoms at a two-year follow-up. Although these symptoms did not predict weight loss at two years, preoperative psychopathology emerged as a signal for persistent mental health risks, underscoring the importance of pre and postoperative psychosocial monitoring and the availability of adjunctive intervention resources. The same applies to psychological aspects, such as loss of control (LOC) or overeating [37].

A retrospective case-control study investigating suicidal ideation (SI) and behaviors (SB) in a group of 206 adolescents receiving bariatric surgery reported 31 study participants (15%) with current/lifetime SI/SB or event at baseline or occurring in the program, supporting the need for evaluation and monitoring for risk of SI/SB in this group [48]. Interestingly, these authors compared subjects with SI/SB to gender, age, and surgery-type-matched subjects without a lifetime history of SI/SB or psychiatric treatments. From this comparison, adolescents with SI/SB had a significantly lower health-related quality of life and a higher burden of depressive symptoms [48].

More recently, a multicenter study investigating suicidal thoughts and behaviors (STBs) in adolescents who underwent bariatric surgery compared to a non-surgical group did not find a role of BS in increasing/reducing the risk for STBs across the initial four years after surgery [15]. Groups did not differ with regard to current or past ideation, plan, or suicide attempts (n = 18 surgical [16%] vs. n = 10 nonsurgical [18%]; *p* = 0.90). Of interest and consistent with previous studies, weight-surgery-specific factors (weight loss, weight satisfaction) were nonsignificant predictors of STBs. Significant post-surgery correlates at year four included lower quality of life, greater depressive symptomatology, greater externalizing symptomatology, greater dysregulation, illicit drug use in the past year, and exposure to someone who attempted or completed suicide, with hazardous alcohol consumption marginally significant. Of note, STBs and other risky behaviors (i.e., illicit drug use, hazardous alcohol use, or psychological dysregulation) co-occurred in the study sample, consistent with the general adolescent and young adult population. Post hoc review indicated approximately half of those who engaged in STBs post-surgery also reported illicit drug use and/or hazardous alcohol use [15].

In conclusion, while current data pose bariatric surgery as relatively safe and beneficial in terms of weight loss and resolution of related comorbidities, it is also important to assess long-term outcomes in a broader way, with a focus on the fact that adolescence is an age where the body, brain, and mind are going through several developmental changes. It is important to consider the relative benefits to each individual patient and to assess the tailored presurgical and postsurgical support they will need in order to ensure beneficence and nonmaleficence.

### 3.4. Justice

The principle of justice pertains to medical ethics as “a fair and equitable distribution of health resources” [16,28].

Bariatric surgery resulted to be cost-effective [50] and cost-saving in adult patients affected with type 2 diabetes [51]. More recently, significant weight loss and durable improvement in cardiovascular risk factors in type 2 diabetes (T2D) and in hypertension were shown in adolescents undergoing BS [6,52]. Taken together, these data suggest that BS may represent a cost-saving procedure in terms of prevention and treatment of chronic conditions. 

Despite this, it is possible that some obese patients might not have the means to pay for surgery, as this procedure has a relevant immediate cost. This is particularly true in countries where the Health System does not cover, or only partially covers, these costs. This is a very important concern that we need to find a way to mitigate, especially since patients from lower socioeconomic backgrounds are often considered “difficult patients”. They are likely to present with more and worse comorbidities, and, at the same time, the likelihood of failure to adhere to the lifestyle changes that should follow is rather high due to lack of means or poor standards of living; these factors tend to worsen the prognosis of surgical treatment both short- and long-term [20]. While some studies predict that the overall cost and burden of untreated or unmanaged obesity and related comorbidities can be much higher than the cost of the surgical intervention and the related procedures, these statistics cannot be corroborated based on the current data. Moreover, regardless of their accuracy, they may not be enough for individual patients who may or may not be able to afford the imminent costs of the surgery [20,53].

To mitigate this, one could imagine at least two different strategies. On one hand, standardization in terms of health care and inherent costs could be helpful, although reaching this goal on a global scale could pose a number of problems. On the other hand, it would be advisable to start well-designed preventative strategies to fight the rise of obesity, as well as intervention plans tailored for the population groups more frequently affected by obesity. It would be unethical to let non-surgical candidates suffer the burden of obesity without intervention or support, especially since there is still a possible improvement of comorbidities with early non-surgical intervention.

Besides the high cost of surgery, postoperative care is also costly and can be lifelong. Psychological support, tailored food supplements, medications, and lifestyle changes must be weighed against the comorbidities and complications that not undertaking the surgery would present. 

Another aspect to consider is that awareness and sensitivity in dealing with obese patients are not equally distributed around the globe due to socio-economical as well as cultural factors. Stigma against obesity is present even amongst healthcare professionals in many countries [54,55]. This creates a further barrier to access as people are less likely to seek help for fear of being negatively judged or even receiving criticism instead of useful interventions. Doctors’ and patients’ factors, in this case, further limit the chance of a recommendation for bariatric surgery and provide a relevant example of an intangible reality that can violate the principle of justice.

Leaving apart the costs involved, in order to understand discrepancies between patients who need care, patients who seek it, how often they get the care they need, and the reasons for denial, it may be important to assess the current climate and attitudes that the stakeholders possess towards bariatric surgery. This includes physicians, adolescent patients, and parents/guardians and could provide insight into any potential subconscious biases or blatant stigmas that may still persevere with respect to different countries and cultures.

One Dutch study, published in 2021, attempted to gauge this through surveys in order to understand potential biases that could influence decision-making [56]. It revealed that only roughly 60% of pediatricians considered bariatric surgery to be an effective way of managing severe obesity in adolescents upon failure of conventional conservative treatment. A significant proportion of the pediatricians still adhered to the outdated guideline of Tanner staging and wanted to limit inclusion to patients who are 16 or older. The numbers were about the same for parents of obese or overweight adolescents, while almost 74% of adolescents agreed that surgery should be made available. The study concluded that there was a positive ongoing change in the acceptance of bariatric surgery as an intervention, although this barely reached the majority when it came to the stakeholders involved in the decision-making [56].

In conclusion, in terms of justice, several questions still need to be answered in order to obtain an accurate determination of the cost-benefit analysis.

## 4. Discussion

Current data support the clinical beneficence of bariatric surgery in adolescents in terms of weight loss and resolution of comorbidities [3,6]. However, more studies need to be performed exploring samples with different ethnical, cultural, and geographical characteristics. Moreover, in order to ensure nonmaleficence, it would be useful to have more data concerning the psychosocial outcomes of the surgery, since papers dealing with these issues are comparatively fewer in number than those regarding the clinical outcomes with respect to weight loss and physical comorbidities.

Apart from subconscious biases that may influence decision-making, the resistance against referrals for bariatric surgery for adolescent obese patients could also be reinforced because, despite the clinical guidelines for eligibility, there is no accepted standardization in terms of screening evaluations, especially pertaining to the ethical aspects. A recent paper [19] proposes the development of a framework for ethical consultations that could alleviate the burden of this decision-making process. It quotes two case studies and analyses the clinical qualms that emerged and the ethical principles that could be at risk based on the decision made.

Although well-designed studies following evidence-based rules are needed, one should also consider the possible contribution offered by more subjective and qualitative data, as they could help physicians to better understand what life after a bariatric surgery procedure can look like. Data regarding the intrinsic friction of the healthcare community with regard to referring adolescent patients for an evaluation for bariatric surgery may also need to be assessed in order to tackle any potential biases that could cause an unjustified deviation toward a lack of referrals. This could also be relevant because recent data support the risk of a “late overcompensation”, the referral for bariatric surgery of patients not responding to other interventions only after a long trial of these approaches, i.e., with a timing that is not appropriate to the existing evidence suggesting better outcomes for earlier surgery [30].

Additionally, the consideration for better access to health care and the effort to improve the patient’s status cannot be limited only to surgical candidates but must be a priority when treating any patient seeking an intervention.

It is important to recognize that with or without the use of bariatric surgery, there has to be an improvement in the way we treat the young obese population. There has to be increased awareness about the sensitivity needed for optimal delivery of clinical care, with a person-specific plan which could include medical and psychological interventions as well as bariatric surgery following balanced and rational strategies. This could require better dissemination of knowledge regarding the multifaceted etiology of obesity, not only among healthcare professionals, but among all stakeholders. One can hope that this will help reduce the overarching stigma associated with the condition; this, in turn, could allow those who may not be eligible for surgery or may not be able to imminently bear all the costs associated with it, to have a better chance at living lives that are less burdened by marginalization, poor self-worth, shame, and guilt.

Especially now that we are aware of the reciprocal effects of psychopathologies and obesity, it is important to tackle them both together. It might be beneficial to work on improving the patient’s mental and social well-being in general, in order to better achieve potential weight-loss goals. Interventions that focus solely on eating habits and physical activity may still be operating under the clearly outdated notion that obesity is simply a result of excess intake of calories and inadequate energy output. Focusing on the overall health of an adolescent patient may instead allow the patient to feel understood and to establish a relationship of trust in the medical staff that facilitate the access to the underlying psychosocial or somatic issues involved in the etiopathogenesis of obesity. This could aid in planning all adequate interventions (for example, implementing healthier coping strategies) that may need to be considered in order to offer long-term support that is patient-centered and not weight-loss centered. It is important to remind ourselves that we are treating the patient and not their obesity, a principle that can, of course, be extended to those who have undergone surgery as well and are potentially struggling to meet their goals.

It is imperative that we work on these tools as soon as possible because the surge in adolescents seeking healthcare for their obesity has increased during the SARS-CoV-2 crisis, and an obese status could pose a greater risk to COVID-related hospitalizations and negative global prognosis [2,57].

## 5. Conclusions

A wider dissemination of scientific data concerning bariatric surgery in children and adolescents is highly important from an ethical perspective. Having clear data concerning indications, risks, and contraindications of the different techniques is fundamental for the physician to provide a better-tailored suggestion, optimizing beneficence and nonmaleficence towards the patient. At the same time, in order to be respected in his/her autonomy, the patient needs to have clear and scientifically sound information to choose whether or not to accept the treatments suggested by his/her doctor. This is fundamental to express valid informed consent to these procedures (and to any other intervention). While the caution around the adoption of bariatric surgery as an intervention to treat adolescents with obesity is valid, given the lack of conclusive data, this should not become a crutch for subconscious biases that unjustly preclude patients from the care they need. To mitigate this, extra data can help with the construction of an ethical framework for the eligibility evaluations of adolescent bariatric candidates [5,21]. Future reviews might be able to benefit from these studies and could apply a standardized methodology (e.g., PRISMA) to overcome a limitation of our work.

Lastly, bariatric surgery should be continually improved to ensure the best safety profile for the candidates [58]. However, the focus on surgical improvement should not eclipse continued research for management that is less radical and applicable at more affordable costs, especially since obesity is not limited to developed countries. Additionally, these ventures need to happen alongside the application of better preventative strategies and enhanced ability to offer better and holistic conservative treatment when necessary.

## Data Availability

Not applicable.

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
