# Peer review of "Ethics of Bariatric Surgery in Adolescence and Its Implications for Clinical Practice"

_ijerph, 2023, doi:10.3390/ijerph20021232_

Round 1

Reviewer 1 Report

1)The paper is a narrative review about an important topic -  mainly the ethical concerns that surround BS in the adolescent population. The methodology used for this review could be stated more formally - in other words, did one author review the papers, or both....was there a third author to "resolve any disputes" about claims in a paper?  There were 7 authors listed....did they all review the papers or just some?? How many papers ere found and reviewed for this review?? Beyond the keywords, did the authors use any system such as PRISMA to establish criteria for including articles in the review? Were MeSH terms used to expand terms such as bariatric surgery to include all types, not just Gastric Sleeve?  

2) Some minor grammar issues - including that many "paragraphs" were single sentences.  Also, several run-on sentences throughout paper.  The aim of the review may be clearer if written as primary and secondary aim (primary =ethical; secondary = improve (or maybe inform!) clinical practice.

3)First paragraph under autonomy - the final sentence "It is intuitive...." is not clearly stated.  

4) Typo line 107 (eventually)

5)Sentence line 109 is not clear - if the child has sound cognitive and emotional ability.... then why do you think they do not have the capacity to consent /  assent?  Is almost like a double negative as you have stated it.... consider revising this section.  Stating that BS causes many long term changes  that may not be clearly understood by an adolescent may suffice!

6) Lines 131-33 are not clear

7)The section on IDD should also include Autism Spectrum Disorder (ASD)

8) Beneficence section is not clearly written and disjointed. Is the goal here to promote one technique over another?

9) Section 3.3 is well written, and clearly stated. 

10) Justice section: statistic on BS and cost savings with diabetes....ref is adults - right?  That is fine, but may need to state that  study was longitudinal on adults  and then circle back to the implications for cost savings for adolescent and how it impacts the prevention of onset of type 2 diabetes....

11) First line of discussion -is current data regarding clinical beneficence "substantial"??

12) Not sure what line 370-371 means?
13) Lines 412-413 regarding Covid could also be included in introduction as the justification for doing this review.

14) line 418 is not a complete sentence

15) Line 419  - who it "it"?? Suggest revision.

16) First paragraph of conclusion needs revision

Reviewer 2 Report

Given the increasing prevalence of obesity in children and adolescents as well, the topic presented in the paper is very important. However, the authors need to cite 3 important source texts. 

Writing: "According to the American Society of Metabolic and Bariatric Surgery (ASMBS) guidelines BS is proposed in adolescents with BMI ≥ 35 kg/m2 (moderate obesity) with major 53 comorbidities or with a BMI ≥ 40 kg/m2 (severe obesity) with minor comorbidities [4,7]" they should cite the source: Mechanick JI, Apovian C, Brethauer S, Garvey WT, Joffe AM, Kim J, Kushner RF, Lindquist R, Pessah-Pollack R, Seger J, Urman RD, Adams S, Cleek JB, Correa R, Figaro MK, Flanders K, Grams J, Hurley DL, Kothari S, Seger MV, Still CD. Surg Obes Relat Dis. 2020 Feb;16(2):175-247; Mechanick JI, Kushner RF, Sugerman HJ, Gonzalez-Campoy JM, Collazo-Clavell ML, Spitz AF, Apovian CM, Livingston EH, Brolin R, Sarwer DB, Anderson WA, Dixon J, Guven S; American Association of Clinical Endocrinologists; Obesity Society; American Society for Metabolic & Bariatric Surgery.Obesity (Silver Spring). 2009 Apr;17 Suppl 1:S1-70

In addition, new guidelines should be included in the paper, which also provide recommendations for qualifying children and adolescents 

Eisenberg D, Shikora SA, Aarts E, Aminian A, Angrisani L, Cohen RV, de Luca M, Faria SL, Goodpaster KPS, Haddad A, Himpens JM, Kow L, Kurian M, Loi K, Mahawar K, Nimeri A, O'Kane M, Papasavas PK, Ponce J, Pratt JSA, Rogers AM, Steele KE, Suter M, Kothari SN. Obes Surg. 2022 Nov 7. doi: 10.1007/s11695-022-06332-1. Online ahead of print. PMID: 36336720

Round 2

Reviewer 1 Report

 Manuscripts reads better however still some typos and grammar issues that need to be addressed. Can you put the work through spell check and grammar check???

Author Response

We thank the reviewer for the appreciation of our work and for his/her suggestions.

We performed a thorough check of grammar and spelling.